# A New Belief Entropy Based on Deng Entropy

**DOI:** 10.3390/e21100987

**Published:** 2019-10-10

**Authors:** Dan Wang, Jiale Gao, Daijun Wei

**Affiliations:** School of Science, Hubei Minzu University, Enshi 445000, China; 201612798@hbmy.edu.cn (D.W.); 201611158@hbmy.edu.cn (J.G.)

**Keywords:** uncertainty measure, Dempster–Shafer evidence theory, Shannon entropy, Deng entropy

## Abstract

For Dempster–Shafer evidence theory, how to measure the uncertainty of basic probability assignment (BPA) is still an open question. Deng entropy is one of the methods for measuring the uncertainty of Dempster–Shafer evidence. Recently, some limitations of Deng entropy theory are found. For overcoming these limitations, some modified theories are given based on Deng entropy. However, only one special situation is considered in each theory method. In this paper, a unified form of the belief entropy is proposed on the basis of Deng entropy. In the new proposed method, the scale of the frame of discernment (FOD) and the relative scale of a focal element with reference to FOD are considered. Meanwhile, for an example, some properties of the belief entropy are obtained based on a special situation of a unified form. Some numerical examples are illustrated to show the efficiency and accuracy of the proposed belief entropy.

## 1. Introduction

Recently, Dempster–Shafer evidence theory [1,2] has been a useful theory for dealing with uncertainty information. It has a wide range of applications and has been extensively studied in many fields, such as decision-making [3,4,5,6], supplier management [7,8,9], pattern recognition [10], risk evaluation [11,12,13], probability density estimation [14], complex network [15,16], and so on. Dempster–Shafer evidence theory is regarded as an extension of the Bayesian theory. The basic probability assignment (BPA) is a fundamental concept in Dempster–Shafer evidence theory. Each BPA represents how strongly the evidence supports one of the elements of the frame. By giving a confidence degree for any subsets of the system, the information of quantitative or qualitative formation is represented by BPA. Meanwhile, some pieces of evidence can be combined into one piece of evidence by the rule of Dempster’s combination. However, there are some open issues in Dempster–Shafer evidence theory. For examples, counterintuitive results may be obtained from some highly conflicting evidence [2,17]. For real applications, a collectively exhaustive and mutually exclusive set in FOD is difficult to be satisfied for real applications [18,19]. Uncertainty modeling is usually discussed when we deal with these open issues. Therefore, how to manage the uncertainty of (BPA) accurately and efficiently is significant and has attracted widespread attention [20,21]. Some methods are proposed for measuring the uncertainty of Dempster–Shafer evidence theory such as Yager’s dissonance measures [22], distance-based measure [23,24], weighted Hartley entropy [25], Klir and Ramer’s discord measure [26] and George and Pal’s conflict measure [27].

Shannon entropy is an effective measure to handle the uncertainty of the system. Although Shannon entropy was first developed to model an uncertain measure of information volume in information theory, it is widely applied for measuring the uncertainty of kinds of systems and processes [28,29]. However, for measuring the uncertainty of Dempster–Shafer evidence theory, Shannon entropy theory can’t be applied because a mass function is a generalized probability, which is assigned on the power set of FOD in Dempster–Shafer evidence theory [30]. In order to address this problem, some modified entropy theories are proposed based on Shannon entropy such as Yager’s dissonance measures, distance-based measure, weighted Hartley entropy [22,24,25,27,31]. However, for some cases, these entropy theories can’t effectively measure the uncertainty of Dempster–Shafer evidence theory [32].

Based on Shannon entropy, a new entropy, known as Deng entropy, has been proposed recently [32]. Deng entropy uses the BPA of the evidence and the cardinality of the element of the BPA as variables to calculate the uncertainty of evidence. That is, Deng entropy considered not only the BPA of the evidence, but also the cardinality of the element of the BPA. Therefore, Deng entropy has successfully solved many practical applications and been applied to many fields [33,34,35]. Meanwhile, Deng entropy can degenerate into the Shannon entropy when the cardinality of elements in BPA is 1. Recently, some limitations of Deng entropy have been found. Deng entropy only considers the BPA of the evidence and the cardinality of the element of the BPA, without the scale of FOD. In fact, the scale of FOD is an important factor for measuring uncertainty of evidence theory [30]. Therefore, some modified methods are proposed for overcoming these limitations of Deng entropy. For examples, Zhou et al. [36], Pan et al. [37] and Cui et al. [38] modified Deng entropy, respectively. In three references, they all consider the scale of FOD and the relative scale between a focal element of FOD with itself. In these references, the relative scale of these is represented from different views, respectively. Therefore, although these methods take into consideration of the scale of FOD and the influence of the intersection between statements on uncertainty, each method only considers one special situation. In this paper, a unified form about belief entropy based on Deng entropy is proposed. The proposed entropy can improve the performance of Deng entropy by considering the scale of the FOD and the relative scale of a focal element with respect to FOD. Meanwhile, the proposed method keeps all the benefits of Deng entropy, and the proposed method can also degenerate into Shannon entropy in the sense of the probability consistency. Some numerical examples are used to illustrate the effectiveness of the proposed entropy.

This paper is organized as follows. In Section 2, some concepts about Dempster–Shafer evidence theory, Shannon entropy, Deng theory and some uncertainty measures in the Dempster–Shafer framework are briefly introduced. In Section 3, the new belief entropy is presented. In Section 4, some numerical examples are given to verify the validity, as well as a comparative study between the new belief entropy and some other uncertainty measures. The conclusions are given in Section 5.

## 2. Preliminaries

In this section, some methods of uncertainty measurement are briefly introduced, including Dempster–Shafer evidence theory [1,2], Shannon entropy [28], Deng entropy [32] and some other typical uncertainty measures in the Dempster–Shafer framework.

### 2.1. Dempster–Shafer Evidence Theory

Let X be a set of mutually exclusive and collectively exhaustive events, indicated by
(1)$$X=\left\{\theta_{1},\theta_{2},\cdots ,\theta_{i},\cdots ,\theta_{\left|X\right|}\right\},$$
where X is an FOD, and $2^{X}$ is the power set of X; we have:(2)$$2^{X}=\left\{\Phi ,\left\{\theta_{1}\right\},\cdots ,\left\{\theta_{\left|X\right|}\right\},\left\{\theta_{1},\theta_{2}\right\},\cdots \left\{\theta_{1},\theta_{2},\cdots \theta_{i}\right\},\cdots ,X\right\}.$$

A mass function, denoted as BPA, is defined as a mapping of the power set $2^{X}$ to the interval [0,1]:(3)$$m:2^{X}\to \left[0,1\right],$$
where the mass function satisfies the following conditions:(4)$$m\left(\Phi \right)=0,\sum_{A\in 2^{X}}m(A)=1,$$
where m(A) represents the belief degree to *A*, namely the degree of evidence supports *A*.

A BPA can also be represented by its associated belief function (Bel) and plausibility function (Pl), respectively. Bel and Pl are defined as follows: respectively:(5)$$\mathrm{Bel}(A)=\sum_{B\subseteq A\neq \Phi }m(A),Pl\left(A\right)=1-Bel\left(\bar{A}\right)=\sum_{B\cap A\neq \Phi }m(B).$$

In Dempster–Shafer evidence theory, there are two BPAs indicated by $m_{1}$ and $m_{2}$, respectively. They can be combined by Dempster’s rule of combination as follows:(6)$$m\left(A\right)=\left\{\begin{array}{ll} \frac{1}{1-k}\sum_{B\cap C=A}m_{1}\left(B\right)m_{2}\left(C\right) & A\neq \Phi ,\\ 0 & A=\Phi , \end{array}\right.$$
where *k* is a normalization constant, which represents the degree of conflict between $m_{1}$ and $m_{2}$; *k* is defined as follows:(7)$$k=\sum_{B\cap C=\Phi }m_{1}\left(B\right)m_{2}\left(C\right),$$
where k<1.

### 2.2. Shannon Entropy

In information theory, Shannon entropy, is an uncertain measure of information volume in a system or process, which is the quantification of the expected value of the information in a message. Shannon entropy, which is denoted as *H*, is defined as follows [28]:(8)$$H=-\sum_{i=1}^{N}p_{i}\log_{a}p_{i},$$
where *N* is the number of basic states, and $p_{i}$ is the probability of state *i*. We have $\sum_{i=1}^{N}p_{i}=1$. Usually, a=2, which means that the unit of information is bits.

Although Shannon entropy is first developed to model an uncertain measure of information volume in information theory, there are still some limitations for measuring the uncertainty of Dempster–Shafer evidence theory [38]. Therefore, other entropies are given for measuring the uncertainty of Dempster–Shafer evidence theory. In the next section, some uncertainty measures about Dempster–Shafer framework are introduced.

### 2.3. Some Uncertainty Measures for Dempster–Shafer Framework

*X* is an FOD. There are focal elements of the mass function of *X*, which are called *A* and *B*, and |A| denotes the cardinality of *A*. For the Dempster–Shafer method, some definitions of uncertain measures are briefly introduced as follows.

#### 2.3.1. Hohle’s Confusion Measure

Hohle’s confusion measure, denoted as $C_{H}$, is defined as follows [26]:(9)$$C_{H}\left(m\right)=-\sum_{A\subseteq X}m\left(A\right)\log_{2}Bel\left(A\right).$$

#### 2.3.2. Yager’s Dissonance Measure

Dissonance measure, denoted as $E_{Y}$, is defined as follows [22]:(10)$$E_{Y}\left(m\right)=-\sum_{A\subseteq X}m\left(A\right)\log_{2}Pl\left(A\right).$$

#### 2.3.3. Dubois and Prade’s Weighted Hartley Entropy

Dubois and Prad’s weighted Hartley entropy, denoted as $E_{DP}$, is defined as follows [25]:(11)$$E_{DP}\left(m\right)=-\sum_{A\subseteq X}m\left(A\right)\log_{2}\left|A\right|.$$

#### 2.3.4. Klir and Ramer’s Discord Measure

Another discord measure, denoted as $D_{KR}$, is defined as follows [26]:(12)$$D_{KR}\left(m\right)=-\sum_{A\subseteq X}m\left(A\right)\log_{2}\sum_{B\subseteq X}m(B)\frac{\left|A\cap B\right|}{\left|B\right|}.$$

#### 2.3.5. Klir and Parviz’s Strife Measure

Klir and Parviz’s strife measure, denoted as $S_{KP}$, is defined as follows [39]:(13)$$S_{Kp}\left(m\right)=-\sum_{A\subseteq X}m\left(A\right)\log_{2}\sum_{B\subseteq X}m(B)\frac{\left|A\cap B\right|}{\left|A\right|}.$$

#### 2.3.6. George and Pal’s Conflict Measure

The total conflict measure proposed by George and Pal, denoted as $TC_{GP}$, is defined as follows [27]:(14)$$TC_{GP}\left(m\right)=\sum_{A\subseteq X}m(A)\sum_{B\subseteq X}m(B)\left(1-\frac{\left|A\cap B\right|}{\left|A\cup B\right|}\right).$$

### 2.4. Deng Entropy and Its Modified Entropy

As a generalization of Shannon entropy, Deng entropy is given as follows:(15)$$E_{d}=-\sum_{A\subseteq X}m\left(A\right)\log_{2}\frac{m(A)}{2^{\left|\left.A\right|\right.}-1},$$
where *X* is the FOD, *A* represents a proposition in mass function *m* and |A| is the cardinality of proposition *A*. From the above definition, Deng entropy can be degenerated to the Shannon entropy if and only if the mass value is assigned to singleton elements, which is $E_{d}=-\sum_{A\subseteq X}m\left(A\right)\log_{2}m\left(A\right)$.

#### 2.4.1. Zhou et al.’s Entropy

Zhou et al. considered the scale of FOD, and defined another belief entropy as follows [30]:(16)$$E_{Md}\left(m\right)=-\sum_{A\subseteq X}m\left(A\right)\log_{2}\left(\frac{m(A)}{2^{\left|A\right|}-1}e^{\frac{\left|A\right|-1}{\left|X\right|}}\right),$$
where |A| represents the number of proposition *A*, and |X| represents the cardinality of *X*, which is the FOD.

#### 2.4.2. Pan et al.’s Entropy

Inspired by Deng entropy, Pan et al. give an entropy base on the probability interval [Bel(A),Pl(A)] in Dempster–Shafer evidence theory. Pan et al.’s entropy is given as follows [37]:(17)$$P_{Bel}\left(m\right)=-\sum_{A\subseteq 2^{\theta }}\frac{Bel(A)+Pl(A)}{2}\log_{2}\left(\frac{Bel(A)+Pl(A)}{2^{\left|A\right|}-1}\right).$$

#### 2.4.3. Cui et al.’s Entropy

Based on Deng entropy and Zhou et al.’s entropy, a new belief entropy (Cui et al.’s entropy) is defined as follows [38]:(18)$$E\left(m\right)=-\sum_{A\subseteq X}m\left(A\right)\log_{2}\left(\frac{m(A)}{2^{\left|A\right|}-1}e^{\sum_{B\subseteq X,B\neq A}\frac{\left|A\cap B\right|}{2^{\left|X\right|}-1}}\right).$$

## 3. The Proposed Method

In the framework of Dempster–Shafer evidence theory, the uncertain information can be modeled not only as a mass function, but also as the source of uncertainty and the number of elements in the FOD. Deng entropy only takes the BPA of the evidence and the cardinality of the element of the BPA into consideration; the relative scale of a focal element with respect to FOD and the scale of FOD are totally ignored. Although these factors are considered in the modified method such as in Refs. [30,37,38], they are not a unified form. In this paper, a new belief entropy, which is called W entropy, is proposed. The mass function, source of uncertainty and the scale of FOD are considered in the proposed method. Meanwhile, a unified form about the scale of FOD and the relative scale of a focal element with reference to FOD is given in the proposed method. W entropy is defined as follows:(19)$$E_{W}\left(m\right)=-\sum_{A\subseteq X,A\neq \Phi }m\left(A\right)\log_{2}\left(\frac{m(A)}{2^{\left|A\right|}-1}{\left(1+\varepsilon \right)}^{f(|X|)}\right),$$
where ε is a constant and ε≥0, f|X| is the function about the cardinality of *X*. *A* is a proposition in mass function m, and |A| denotes the cardinality of proposition *A*. From Equation (Equation 19), Deng entropy and its modified entropy are a special case of W entropy. For example, W entropy is the same as Deng entropy when ε=0 in Equation (Equation 19). For Zhou et al.’s entropy (Equation (Equation 16)) and Cui et al.’s entropy (Equation (Equation 18)), they are ε=e−1, $f\left(|X|\right)=\left|A\right|-1$ and $f\left(|X|\right)=\sum_{B\subseteq X,B\neq A}\frac{\left|A\cap B\right|}{2^{\left|X\right|}-1}$ in W entropy (Equation (Equation 19)), respectively. That is, ε is an indeterminate nonnegative number, and it can take an appropriate number based on the actual example. In fact, compared with Zhou et al.’s entropy and Cui et al.’s entropy, it is more typical to replace 1+ε with *e*.

Meanwhile, when the number of conditionals of that increases by the scale of the frame of discernment in Zhou et al.’s entropy and Cui et al.’s entropy, the influence of the number of conditions for the intersection of elements will be greatly reduced for the measurement of the uncertainty of the evidence. Therefore, in this paper, let $f\left(|X|\right)=\sum_{\underset{B\neq A}{B\subseteq X}}\frac{\left|A\cap B\right|}{\left|A\cup B\right|}$, and we have W entropy that is denoted as follows:(20)$$E_{W}^{{}^{\prime }}\left(m\right)=-\sum_{A\subseteq X,A\neq \Phi }m\left(A\right)\log_{2}\left(\frac{m(A)}{2^{\left|A\right|}-1}{\left(1+\varepsilon \right)}^{\sum_{\underset{B\neq A}{B\subseteq X}}\frac{\left|A\cap B\right|}{\left|A\cup B\right|}}\right),$$
where |A∩B| is the cardinality of the intersection of *A* and *B*, and |A∪B| is the cardinality of the union set of *A* and *B*.

## 4. Numerical Examples

In this section, some examples are used to present the effectiveness and superiority of W entropy based on Equation (Equation 20). Firstly, the classical problem of Dempster–Shafer evidence theory is discussed.

In order to solve the high conflict evidence combination problem of D–S evidence, a new method of evidence modification is established and then based on the new entropy in this paper.

Step 1: Through the sensor, we obtain the evidence bodies in different directions and determine their BPA, which are recorded as $m_{1},m_{2},\cdots ,m_{n}$. They represent different decision schemes and have some influence on the final decision-making scheme.

Step 2: With the new proposed entropy (W entropy), determine the entropy of each piece of evidence, denoted as $E_{w}\left(m_{1}\right),E_{w}\left(m_{2}\right),\cdots ,E_{w}\left(m_{n}\right).$

Step 3: Intuitively, the more confusing the evidence is, the more uncertain the information that it contains, the accuracy is then lower, and vice versa. Hence, we define the formula of the weight of evidence as follows:$$W\left(m_{i}\right)=\frac{2^{-E_{w}\left(m_{i}\right)}}{\sum_{i=1}^{n}2^{-E_{w}\left(m_{i}\right)}},\left(\sum_{i=1}^{n}2^{-E_{w}\left(m_{i}\right)}\neq 0,i=1,2,\cdots n\right),$$
where $E_{w}\left(m_{i}\right)$ is the entropy value of the evidence $m_{i}$, and $W(m_{i})$ is the weight of the evidence $m_{i}$. When $E_{w}\left(m_{i}\right)=0$, the value of the function is equal to 1, and, when the $E_{w}\left(m_{i}\right)$ gradually increases, the value of the function approaches zero at an exponential rate.

Step 4: By using a combination of D–S evidence theory, the BPA of the modified evidence is combined by n−1 times to obtain the final decision-making scheme.

Example 1: Assume that the FOD is X={A,B}. The results of FOD are presented and listed as follows:$$\begin{array}{c} m_{1:}m_{1}\left(\left\{A\right\}\right)=0.99,m_{1}\left(\left\{B\right\}\right)=0,m_{1}\left(\left\{A,B\right\}\right)=0.01,\\ m_{2:}m_{2}\left(\left\{A\right\}\right)=0,m_{2}\left(\left\{B\right\}\right)=0.99,m_{2}\left(\left\{A,B\right\}\right)=0.01. \end{array}$$

The results of the combination by using the Dempster’s combination rule and the proposed method are shown in Table 1, respectively.

In fact, $m_{1}\left(A\right)=0.99$ and $m_{2}\left(B\right)=0.99$ means that *A* and *B* are strongly supported in the $m_{1}$ and $m_{2}$, respectively. While m(A)=m(B)=0 in the results of a combination using the Dempster’s combination rule. Therefore, from Table 1, the results of Dempster–Shafer evidence theory are counter-intuitive. However, the result (m(A)=m(B)=0.4999) of the proposed method is feasible and resultful.

Secondly, an example is given for comparing the rule of the Deng entropy, Zhou et al.’s entropy and Cui et al.’s entropy, and the proposed method.

Example 2: Assume that the FOD is ${X=\{a}_{1}{,a}_{2},\cdots {,a}_{20}\}$. The results present a body of evidence (BOE) listed as follows:$$m_{1}:m_{1}\left(\left\{a_{1},a_{2},\cdots ,a_{10}\right\}\right)=0.4,m_{1}\left(\left\{a_{10},a_{11},\cdots ,a_{20}\right\}\right)=0.6,$$
$$m_{2}:m_{2}\left(\left\{a_{1},a_{2},\cdots ,a_{10}\right\}\right)=0.4,m_{2}\left(\left\{a_{1},a_{2},\cdots ,a_{5},a_{16},a_{17},\cdots ,a_{20}\right\}\right)=0.6.$$

The uncertainty of $m_{1}$ and $m_{2}$ are used by the Deng entropy, Zhou et al.’s entropy and Cui et al.’s entropy, respectively. The results are as follows:

Deng entropy:$$\begin{array}{c} E_{d}\left(m_{1}\right)=-\sum_{A\subseteq X}m_{1}\left(A\right)\log_{2}\frac{m_{1}\left(A\right)}{2^{\left|A\right|}-1}\\ =-0.4\log_{2}\frac{0.4}{\left(2^{10}-1\right)}-0.6\log_{2}\frac{0.6}{\left(2^{10}-1\right)}=10.9695, \end{array}$$
$$\begin{array}{c} E_{d}\left(m_{2}\right)=-\sum_{A\subseteq X}m_{2}\left(A\right)\log_{2}\frac{m_{2}\left(A\right)}{2^{\left|A\right|}-1}\\ =-0.4\log_{2}\frac{0.4}{\left(2^{10}-1\right)}-0.6\log_{2}\frac{0.6}{\left(2^{10}-1\right)}=10.9695. \end{array}$$

Zhou’s belief entropy:$$\begin{array}{c} E_{Md}\left(m{}_{1}\right)=-\sum_{A\subseteq X}m_{1}\left(A\right)\log_{2}\left(\frac{m{}_{1}\left(A\right)}{2^{\left|A\right|}-1}e^{\frac{\left|A\right|-1}{\left|X\right|}}\right)\\ =-0.4\log_{2}\left(\frac{0.4}{2^{10}-1}e^{{}^{\frac{10-1}{10}}}\right)-0.6\log_{2}\left(\frac{0.6}{2^{10}-1}e^{\frac{10-1}{10}}\right)\\ =9.6711, \end{array}$$
$$\begin{array}{c} E_{Md}\left(m_{2}\right)=-\sum_{A\subseteq X}m_{2}\left(A\right)\log_{2}\left(\frac{m_{2}\left(A\right)}{2^{\left|A\right|}-1}e^{\frac{\left|A\right|-1}{\left|X\right|}}\right)\\ =-0.4\log_{2}\left(\frac{0.4}{2^{10}-1}e^{\frac{10-1}{10}}\right)-0.6\log_{2}\left(\frac{0.6}{2^{10}-1}e^{\frac{10-1}{10}}\right)\\ =9.6711. \end{array}$$

Cui et al.’s entropy:$$\begin{array}{c} E\left(m_{1}\right)=-\sum_{A\subseteq X}m_{1}\left(A\right)\log_{2}\left(\frac{m_{1}\left(A\right)}{2^{\left|A\right|}-1}e^{\sum_{\underset{B\neq A}{B\subseteq X}}\frac{\left|A\cap B\right|}{2^{\left|X\right|}-1}}\right)\\ =-0.4\log_{2}\left(\frac{0.4}{2^{10}-1}e^{0}\right)-0.6\log_{2}\left(\frac{0.6}{2^{10}-1}e^{0}\right)\\ =10.9695, \end{array}$$
$$\begin{array}{c} E\left(m_{2}\right)=-\sum_{A\subseteq X}m_{2}\left(A\right)\log_{2}\left(\frac{m_{2}\left(A\right)}{2^{\left|A\right|}-1}e^{\sum_{\underset{B\neq A}{B\subseteq X}}\frac{\left|A\cap B\right|}{2^{\left|X\right|}-1}}\right)\\ =-0.4\log_{2}\left(\frac{0.4}{2^{10}-1}e^{{}^{\frac{5}{2^{20}-1}}}\right)-0.6\log_{2}\left(\frac{0.6}{2^{10}-1}e^{\frac{5}{2^{20}-1}}\right)\\ =10.9695. \end{array}$$

From these results, these entropies have the same shortages that couldn’t measure the differences of uncertain degree between two BOEs. Deng entropy and Zhou’s belief entropy only considered the two effects of BPAs and the conditional number of focal elements. Cui et al.’s entropy considers the condition number of the intersection between focal elements of evidence on the basis of Deng entropy. However, when the number of conditions for identifying the frame is large, the influence factor of the number of conditions for the intersection of elements between the evidence will be greatly reduced for the measurement of the uncertainty of the evidence. In other words, in the framework of multi-element recognition, Cui et al.’s entropy and Deng entropy have strong similarity. However, the uncertainty of $m_{1}$ and $m_{2}$ is distinguished by the proposed method. According to Equation (Equation 20), two BOEs are calculated and shown as follows:$$\begin{array}{c} W\left(m_{1}\right)=-\sum_{A\subseteq X,A\neq \Phi }m_{1}\left(A\right)\log_{2}\left(\frac{m_{1}\left(A\right)}{2^{\left|A\right|}-1}{\left(1+\varepsilon \right)}^{\sum_{\underset{B\neq A}{B\subseteq X}}\frac{\left|A\cap B\right|}{\left|A\cup B\right|}}\right)\\ =-0.4\log_{2}\left(\frac{0.4}{2^{10}-1}2^{0}\right)-0.6\log_{2}\left(\frac{0.6}{2^{10}-1}2^{0}\right)\\ =10.9695, \end{array}$$
$$\begin{array}{c} W\left(m_{2}\right)=-\sum_{A\subseteq X,A\neq \Phi }m_{2}\left(A\right)\log_{2}\left(\frac{m_{2}\left(A\right)}{2^{\left|A\right|}-1}{\left(1+\varepsilon \right)}^{\sum_{\underset{B\neq A}{B\subseteq X}}\frac{\left|A\cap B\right|}{\left|A\cup B\right|}}\right)\\ =-0.4\log_{2}\left(\frac{0.4}{2^{10}-1}2^{\frac{5}{20}}\right)-0.6\log_{2}\left(\frac{0.6}{2^{10}-1}2^{\frac{5}{20}}\right)\\ =10.7195. \end{array}$$

From these results, W entropy not only considered the scale of the FOD and the influence of the intersection between statements on uncertainty in BPA, but also solved the problem of uncertain measurement under the framework of multi-element recognition.

Example 3 [30]: Given a frame of discernment *X* with 15 elements that are denoted as element 1, element 2, etc. That is, the FOD X={1,2,3,…,15}. A mass function is shown as follows:m({3,4,5})=0.05,m({6})=0.05,m(A)=0.8,m(X)=0.1,
where *A* is a variable subset, the number of the element of *A* is changed from element 1 to element 14. When the *A* changes, Deng entropy, Cui’s entropy and W entropy are calculated and shown in Table 2 and Figure 1, respectively.

From Table 2 and Figure 1, the results showed that some other entropies and our proposed method are smaller than Deng entropy and Cui’s entropy. This is reasonable because more information in the BOE is taken into consideration within these entropies and the proposed method. When the frame has fewer elements, the W entropy gets almost the same result as the calculation of Cui’s entropy. In particular, when the mass value (BPA) is assigned only on a singleton element subset and the intersection of focal elements of evidence as empty; W entropy can degenerate to the following equation:$$\begin{array}{c} E_{W}\left(m\right)=-\sum_{A\subseteq X,A\neq \Phi }m\left(A\right)\log_{2}\left(\frac{m(A)}{2^{\left|A\right|}-1}{\left(1+\varepsilon \right)}^{\sum_{\underset{B\neq A}{B\subseteq X}}\frac{\left|A\cap B\right|}{\left|A\cup B\right|}}\right)\\ =-\sum_{A\subseteq X}m(A)\log_{2}\frac{m(A)}{2^{\left|A\right|}-1}\\ =-\sum_{A\subseteq X}m(A)\log_{2}m\left(A\right). \end{array}$$

As shown in Table 2 and Figure 1, the uncertain degree measured by Deng entropy, Cui’s entropy and the proposed entropy are obviously increased with the linear increasing of |A|. From Figure 1, Deng entropy and Cui’s entropy almost coincide and can’t distinguish the different uncertain degree of them. The proposed entropy (W entropy) successfully solved the above problems, and ensure it to be more reasonable and effective for uncertainty measure in Dempster–Shafer framework.

Meanwhile, some models of W entropy are given on the basis of Equation (Equation 20). The relationship of the value of W entropy and parameter ε is shown in Figure 2.

From Figure 2, W entropy is the same as Deng entropy when ε=0 in Equation (Equation 20). The growth rate of the value of entropy is bigger, the value of ε is smaller. For these issues, the changing of Deng entropy is the fastest. In fact, there is always ${\left(1+\varepsilon \right)}^{\sum_{\underset{B\neq A}{B\subseteq X}}\frac{\left|A\cap B\right|}{\left|A\cup B\right|}}\geq 1$ in Equation (Equation 20).

## 5. Conclusions

Although Deng entropy is a useful method for measuring the uncertainty of BPA in Dempster–Shafer evidence, there are some limitations. Therefore, some entropy theories were modified based on Deng entropy. However, only one special situation is considered in each theory method. This paper represents a unified form of the belief entropy, which considers the scale of the frame of discernment (FOD) and the influence of the intersection between statements on uncertainty. The previous work about Deng entropy is a special situation of the proposed method. Furthermore, when the mass value (BPA) is assigned only on a singleton element subset and the intersection of focal elements of evidence as empty, W entropy can degenerate to Deng entropy and Shannon Entropy. Some numerical examples illustrate the effectiveness and the superiority of the proposed entropy. The proposed entropy gives a unified form of computational uncertainty, which not only considers the scale of the FOD and the influence of the intersection between statements on uncertainty in BPA, but also solved the problem of uncertain measurement under the framework of multi-element recognition.

## Figures and Tables

**Figure 1 entropy-21-00987-f001:**
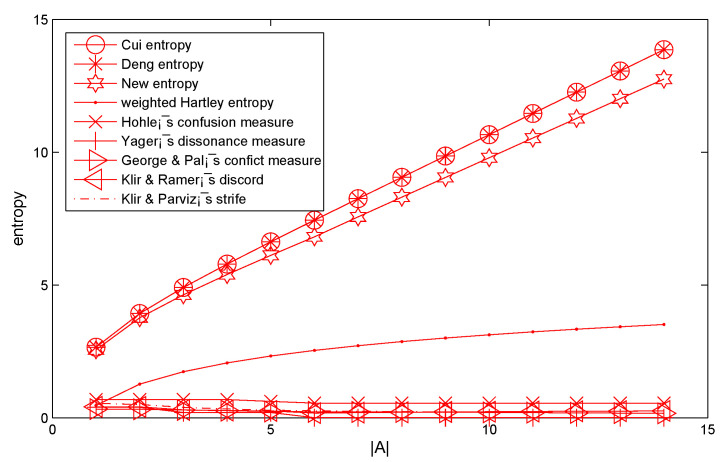
Comparison between the new entropy and other uncertainty measures based on Example 3.

**Figure 2 entropy-21-00987-f002:**
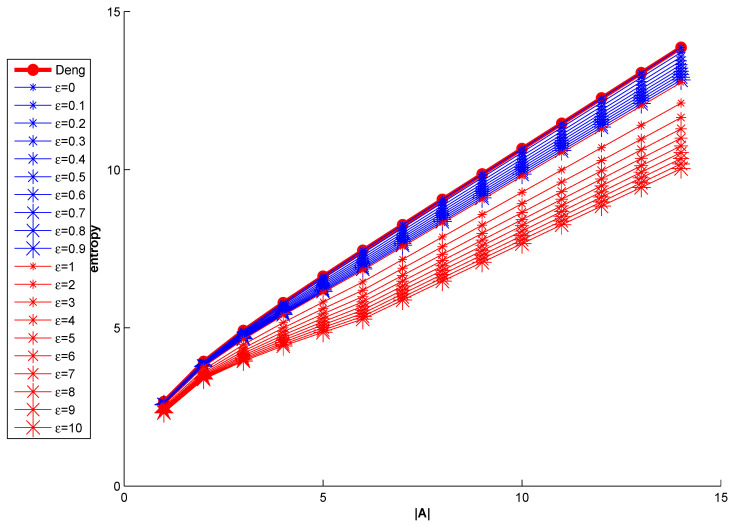
Based on Equation (Equation 20), the relationship of the value of W entropy and parameter ε.

**Table 1 entropy-21-00987-t001:** The results of the new proposed entropy with combination rule.

Fusion Method Focus	A	B	A, B
Dempster’s combination rule	0	0	1
The proposed method	0.4999	0.4999	0.0002

**Table 2 entropy-21-00987-t002:** Deng entropy, Cui’s entropy and W entropy for A changes.

Cases	Deng Entropy	Cui’s Entropy	W Entropy
A={1}	2.6623	2.6622	2.5623
A={1,2}	3.9303	3.9301	3.7703
A={1,2,3}	4.9082	4.908	4.6315
A={1,⋯,4}	5.7878	5.7876	5.3945
A={1,⋯,5}	6.6256	6.6254	6.1156
A={1,⋯,6}	7.4441	7.4438	6.8174
A={1,⋯,7}	8.2532	8.2529	7.5666
A={1,⋯,8}	9.0578	9.0574	8.3111
A={1,⋯,9}	9.8600	9.8596	9.0534
A={1,⋯,10}	10.6612	10.6607	9.7945
A={1,⋯,11}	11.4617	11.4613	10.5351
A={1,⋯,12}	12.2620	12.2615	11.2753
A={1,⋯,13}	13.0622	13.0616	12.0155
A={1,⋯,14}	13.8622	13.8616	12.7556

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
