# Peer review of "A New Belief Entropy Based on Deng Entropy"

_entropy, 2019, doi:10.3390/e21100987_

Round 1

Reviewer 1 Report

As mentioned there are numerous language issues throughout the paper that requires additional editing.

small things like when introducing Shannon, the have N is the number of states then have b=2 for binary systems.  Needs to be changed to N = 2.

in the sentence :... mass value is assigned to single elements... they should say singleton elements not single elements

most importantly, they show a collection of entropies, including their own which is creative, however they fail to provide the equation for how the entropies are used with the BPA combining equation of D-S.  This equation if vital to understand how the entropies are tied to the combining rule.

Author Response

Response to Reviewer #1

Dear reviewer,

Great appreciation for your comments and suggestion as follows. Those comments are valuable and very helpful for revising and improving our paper. We have studied comments carefully and have made corrections which we hope meet with approval. The main corrections in the paper and the responses to your comments are as flowing.

Comments:

As mentioned there are numerous language issues throughout the paper that requires additional editing.

Small things like when introducing Shannon, the have N is the number of states then have b=2 for binary systems. Needs to be changed to N = 2.

In the sentence :... mass value is assigned to single elements... they should say singleton elements not single elements

Most importantly, they show a collection of entropies, including their own which is creative, however they fail to provide the equation for how the entropies are used with the BPA combining equation of D-S. This equation if vital to understand how the entropies are tied to the combining

rule.

Our corresponding revisions are detailed as follows:

Question 1: Language issues throughout the paper that requires additional editing.

Response : Thanks a lot for the reviewer’s comments. These problems do exist in this article, we have major revision in the revised manuscript.

Related Revision:

Page 1:

(1) Line 20: “Meanwhile, BPAs can been combined in the rule of Dempster’s combination” is revised as “Meanwhile, some evidences can be combined into one evidence by the rule of Dempster’s combination”

(2) Line 27: “Some measures are given for measuring the uncertainly of BPA such as……” is revise as “Some methods are proposed for measuring the uncertainty of Dempster-Shafer evidence theory

Page 2:

(3) Line 61: “the preliminaries ……” is revise as “some concepts about Dempster-Shafer evidence theory, Shannon entropy, Deng theory and some uncertainty measures in Dempster-Shafer framework are briefly introduced”.

(4) Line 63: “In section 4, ……” is revise as “In Section 4, some numerical examples are given to verify the validity, as well as a comparative study between the new belief entropy and some other uncertainty measures.

(5) Line 67: “some preliminaries briefly are introduced,” is revise as “some methods of uncertainty measurement are briefly introduced,”

(6) Line 73: change “from” to “of” in the sentence of “A mass function……”is modified as follows:

A mass function is called BPA, is defined as a mapping of the power set to the interval [0,1]:

Page 3:

(7) Line 88: change “b=2” to “a=2” in the sentence of “Usually b=2, which means the unit of information is bit.” is modified as follows:

Usually a=2 which means the unit of information is bit.

(8) Line 91: f “In next section, ……”is modified as follows:

In next section, some uncertainty measures about Dempster-Shafer framework are introduced.

(9) Line 95: “There is a FOD, ……” is revise as “X is a FOD. There are focal elements of the mass function of X,……”.

Page 4:

(9) Line 113: change “single” to “singleton” in the sentence of “Deng entropy ……”is modified as follows:

Deng entropy can be degenerated to the Shannon entropy if and only if the mass value is assigned to singleton elements

(10) Line 116: “where |A| denotes ……” is revise as “where |A| represents the number of proposition A, |X| represents the cardinality of X which is the FOD.”

Page 6:

(11) Line 162: Modify example 1 as follows:

Assuming the FOD is X = {A,B}.The results of FOD are presented by BOEs listed as follows:

(13) Line 168: “Therefore, ……” is revise as “Therefore, from Table 1, the result of Dempster-Shafer evidence theory is counter-intuitive. However, the result m(A)=m(B)=0.4999 in the proposed method is feasible and resultful”.

(14) Line 175: “The results of that are given as follows,” is revise as “The results are as follows.

Page 7:

(15) Line 178: “From the results, ……” is revise as “From these results, these entropies have the same shortages that couldn’t measure the differences of uncertain degree between two BOEs..”

Page 8

(16) Line 196: The sentence of “From Table 1 and Figure 1,” is revise as “From Table 2 and Figure 1,”

(17) Line 202: “As shown in Example 1 and Example 2, ……” is revise as “As shown in Table 2 and Figure 1, the uncertain degree measured by Deng entropy, Cui's entropy and the proposed entropy are increases obviously with the linear increasing of |A|.

(18) Line 203: “However, Deng entropy……” is revise as “From figure 1, Deng entropy and Cui's entropy almost coincide and can’t distinguish the different uncertain degree of them.

Page 9:

(19) Line 207: “Meanwhile, some properties……” is revise as “Meanwhile, some models of W entropy is given on the basis of the equation (20).”

(20) Line 215: “In this paper, ……, of the belief entropy” is revise as “This paper represents a unified form of the belief entropy, which considers the scale of the frame of discernment (FOD) and the influence of the intersection between statements on uncertainty.”

(21) Line 222: “The proposed entropy……” is revise as “The proposed entropy gives a unified form of computational uncertainty, which not only considers. the scale of the FOD and the influence of the intersection between statements on uncertainly in BPA, but also solved the problem of uncertain measurement under the framework of multi-element recognition.”

Page 9:

(19) Line 208: “Meanwhile, some properties……” is revise as “Meanwhile, some models of W entropy are given on the basis of the equation (20).”

(20) Line 215: “In this paper, ……, of the belief entropy” is revise as “This paper represents a unified form of the belief entropy, which considers the scale of the frame of discernment (FOD) and the influence of the intersection between statements on uncertainty.”

(21) Line 222: “The proposed entropy……” is revise as “The proposed entropy gives a unified form of computational uncertainty, which not only considers. the scale of the FOD and the influence of the intersection between statements on uncertainly in BPA, but also solved the problem of uncertain measurement under the framework of multi-element recognition.”

Page 10:

(22) Some references of this paper are revised as follows:

[3] Charnes, A.; Cooper, W.W.; Rhodes, E. Measuring the efficiency of decision making units. European journal of operational research 1978, 2, 429–444.

[4] Bell, D.E. Regret in decision making under uncertainty. Operations research 1982, 30, 961–981.

[5] Polikar, R. Ensemble based systems in decision making. IEEE Circuits and systems magazine 2006, 6, 21–45.

[6] Edwards, W. The theory of decision making. Psychological bulletin 1954, 51, 380.

[10] Hu, M.K. Visual pattern recognition by moment invariants. IRE transactions on information theory 1962,8, 179–187.

[11] Klinke, A.; Renn, O. A New Approach to Risk Evaluation and Management: Risk-Based, Precaution-Based,and Discourse-Based Strategies 1. Risk Analysis: An International Journal 2002, 22, 1071–1094.

Page 11:

[26] Pal, N.R.; Bezdek, J.C.; Hemasinha, R. Uncertainty measures for evidential reasoning I: A review.International Journal of Approximate Reasoning 1992, 7, 165–183.

[39] Florea, M.C.; Grenier, D. Unified approach to the fusion of imperfect data? Proceedings of SPIE – The International Society for Optical Engineering 2002, 4731

Question 2: Small things like when introducing Shannon, the have N is the number of states then have b=2 for binary systems. Needs to be changed to N = 2.

Response : Thank you for the reviewer’s comments. This is a clerical error. When base a is equal to 2, the information is in bits. It be modified as follows:

Related Revision:

Page 3:

Line 84: Just to make it a little bit more, the introduction to Shannon entropy is modified as follows,

In information theory, Shannon entropy, is an uncertain measures of information volume in a system or process, and quantify the expected value of the information contained in a message. Shannon entropy, which denoted as H, is defined as follows

where N is the number of basic states, and is the probability of state i , satisfies
.

Usually a=2, which means the unit of information is bit.

Question 3: In the sentence :... mass value is assigned to single elements... they should say singleton elements not single elements

Response : Thanks a lot for the reviewer’s comments. These errors do occur in the article, which has been corrected as follows.

Related Revision:

Change “single elements” to “singleton elements” in the passage is modified as follows

Page 4:

(1)  Line 114: Deng entropy can be degenerated to the Shannon entropy if and only the mass value is assigned to singleton elements

Page 8:

(2)  Line 203: when the mass value (BPA) is assigned only on singleton element subset

Page 9:

(3)  Line 9: What's more, when the mass value (BPA) is assigned only on singleton element subset and the intersection of focal elements of evidence as empty

Question 4: they fail to provide the equation for how the entropies are used with the BPA combining equation of D-S

Response : Thanks a lot for the reviewer’s comments. It is true that the formula of the entropies and the BPA combining equation of D-S after fusion is not given in this article. Additional edits are as follows:

Related Revision:

Page 5:

An introduction to the fusion method is added before example 1:

In order to solve the high conflict evidence combination problem of D-S evidence,a new method of evidence modification is established and based on the new entropy in this paper.

Step 1:

Through the sensor, we obtain the evidence bodies in different directions, and determine their BPA, which are recorded as. They represent different decision schemes and have some influence on the final decision-making scheme.

Step 2:

With the new proposed entropy (W entropy) determine the entropy of each piece of evidence, denoted as

Step 3:

Intuitively, the more confused evidence is, the more uncertain information it contains, the accuracy is lower, and vice versa. Hence, we define the formula of the weight of evidence as follows:

Where is the entropy value of the evidence , is the weight of the evidence .The reason of choosing this function() as the weight of evidence lies in the rationality of the function. When, the value of the function is equal to 1, and when the gradually increases, the value of the function approaches zero at an exponential rate.

Step 4:

By using combination of D-S evidence theory, the BPA of the modified evidence is combined by times to obtain the final decision-making scheme.

We thank the reviewer for the valuable suggestions to our works. We hope that the deficiencies pointed out in the original submission are overcome in this revised version. If there are any problems in the revised version, please do not hesitate to point out, we will revise the submission according to reviewer’s suggestions.

Reviewer 2 Report

Dear Authors,

in my opinion and after reading the paper I would say that the results you shown are of relevance but it annoys me the poor presentation of these results which confuse me and cover the importance of your study. If these issues were corrected (lack of a more detailed background and not very good use of the English language and style) I will have no doubt to say that this could be a work suited to be published in this journal.

Best regards,

Irene

Author Response

Response to Reviewer #2

Dear reviewer,

Great appreciation for your comments and suggestion as follows. Those comments are valuable and very helpful for revising and improving our paper. We have studied comments carefully and have made corrections which we hope meet with approval. The main corrections in the paper and the responses to your comments are as flowing.

Comments:

In my opinion and after reading the paper I would say that the results you shown are of relevance but it annoys me the poor presentation of these results which confuse me and cover the importance of your study.

If these issues were corrected (lack of a more detailed background and not very good use of the English language and style) I will have no doubt to say that this could be a work suited to be published in this journal.

Our corresponding revisions are detailed as follows:

Question 1: lack of a more detailed background

Response: Thank you very much for your question. Following your question, in introduction of revised manuscript, more detailed background about our proposed method has been introduced.

Related Revision:

In the first paragraph of introduction, some limitations of Dempster-Shafer evidence theory are added as follow,

Recently, Dempster-Shafer evidence theory [1,2] which is a useful theory for deal with uncertainty information……Meanwhile, BPAs can be combined by the rule of Dempster's combination. However, there are some open issues in Dempster-Shafer evidence theory. For examples, counterintuitive results may be obtained from some highly conflicting evidences [2,17]. For real applications, collectively exhaustive and mutually exclusive set in FOD, which is difficult to be satisfied for real applications [18,19]. Uncertainty modeling is usually discussed when we deal with these open issues. Therefore, how to manage the uncertainly of (BPA) accurately and efficiently is of significance and has attract widespread attention [20,21]. ……George and Pal's conflict measure [27].

In the second paragraph of introduction, Shannon entropy is introduced in detail as follows,

Shannon entropy is an effective measure to handle the uncertainly of system. Although Shannon entropy is first developed to model uncertain measure of information volume in information theory, it is widely applied for measuring uncertainty of kinds of systems and process [28,29]. However, for mearing uncertainty of Dempster-Shafer evidence theory, Shannon entropy theory can’t be applied. Because the mass function is a generalized probability, which assigned on the power set of FOD in Dempster-Shafer evidence theory [30]. In order to address this problem, some modified entropy theories are proposed based on Shannon entropy such as Yager's dissonance measures, distance-based measure, weighted Hartley entropy and so on [22,24,25,27,31]. However, for some cases, these entropy theories can’t effectively measure uncertainty of Dempster-Shafer evidence theory [32].

In the third paragraph of introduction, the introduction of Deng entropy and some modified methods is rewritten as follows,

Based on Shannon entropy, a new entropy, which named as Deng entropy, is proposed recently [32]. Deng entropy use the BPA of the evidence and the cardinality of the element of the BPA as variables to calculate the uncertainty of evidence. That is, Deng entropy considered not only the BPA of the evidence, but also the cardinality of the element of the BPA. Therefore, Deng entropy has been successfully solved many practical applications and applied to many fields. Meanwhile, Deng entropy can degenerate into the shannon entropy when the cardinality of elements in BPA is 1. Recently, some limitations of Deng entropy are found. Deng entropy only taking into consider the BPA of the evidence and the cardinality of the element of the BPA, reckon without the scale of FOD. In fact, the scale of FOD is important factor for measuring uncertainty of evidence theory [30]. Therefore, some modified methods are proposed for overcoming these limitations of Deng entropy. For examples, Zhou et al. [36], Pan et al. [37] and Cui et al. [38] modified Deng entropy, respectively. In three references, they all consider the scale of FOD and the relative scale between a focal element of FOD with itself. In these references, the relative scale of that are represented from different views, respectively. Therefore, although these methods take into consideration of the scale of FOD and the influence of the intersection between statements on uncertainly, each method only consider one special situation. In this paper, a unified form about belief entropy of Dempster-Shafer theory based on Deng entropy is proposed. ……

Question 2: not very good use of the English language and style

Response: Thank you very much for your careful reading and provide us the valuable question. We checked our manuscript carefully and have made corrections. For examples, some revisions are shown as follows,

Related Revision:

Page 1:

(1) Line 20: “Meanwhile, BPAs can been combined in the rule of Dempster’s combination” is revised as “Meanwhile, some evidences can be combined into one evidence by the rule of Dempster’s combination”

(2) Line 27: “Some measures are given for measuring the uncertainly of BPA such as……” is revise as “Some methods are proposed for measuring the uncertainty of Dempster-Shafer evidence theory

Page 2:

(3) Line 61: “the preliminaries ……” is revise as “some concepts about Dempster-Shafer evidence theory, Shannon entropy, Deng theory and some uncertainty measures in Dempster-Shafer framework are briefly introduced.

(4) Line 63: “In section 4, ……” is revise as “In Section 4, some numerical examples are given to verify the validity, as well as a comparative study between the new belief entropy and some other uncertainty measures.

(5) Line 67: “some preliminaries briefly are introduced,” is revise as “some methods of uncertainty measurement are briefly introduced,”

(6) Line 73: change “from” to “of” in the sentence of “A mass function……”is modified as follows:

A mass function is called BPA, is defined as a mapping of the power set to the interval [0,1]:’

Page 3:

(7) Line 88: change “b=2” to “a=2” in the sentence of “Usually b=2, which means the unit of information is bit.”is modified as follows:

Usually a=2 which means the unit of information is bit.

(8) Line 91: f “In next section, ……”is modified as follows:

In next section, some uncertainty measures about Dempster-Shafer framework are introduced.

(9) Line 95: “There is a FOD, ……” is revise as “X is a FOD. There are focal elements of the mass function of X……”

Page 4:

(10) Line 113: change “single” to “singleton” in the sentence of “Deng entropy ……”is modified as follows:

Deng entropy can be degenerated to the Shannon entropy if and only if the mass value is assigned to singleton elements

(11) Line 116: “where |A| denotes ……” is revise as “where |A| represents the number of proposition A, |X| represents the cardinality of X which is the FOD.”

Page 6:

(12) Line 162: Modify example 1 as follows:

Assuming the FOD is X = {A,B}. The result of FOD are presented by BOEs listed as follows:

(13) Line 168: “Therefore, ……” is revise as “Therefore, from Table 1, the result of Dempster-Shafer evidence theory is counter-intuitive. However, the result m(A)=m(B)=0.4999 in the proposed method is feasible and resultful.”

(14) Line 175: “The results of that are given as follows,” is revise as “The results are as follows.

Page 7:

(15) Line 178: “From the results, ……” is revise as “From these results, these entropies have the same shortages that couldn’t measure the differences of uncertain degree between two BOEs.”

Page 8:

(16) Line 196: The sentence of “From Table 1 and Figure 1,” is revise as “From Table 2 and Figure 1,”

(17) Line 202: “As shown in Example 1 and Example 2, ……” is revise as “As shown in Table 2 and Figure 1, the uncertain degree measured by Deng entropy, Cui's entropy and the proposed entropy are increases obviously with the linear increasing of |A|.

(18) Line 203: “However, Deng entropy……” is revise as “From figure 1, Deng entropy and Cui's entropy almost coincide and can’t distinguish the different uncertain degree of them.

Page 9:

(19) Line 208: “Meanwhile, some properties……” is revise as “Meanwhile, some models of W entropy are given on the basis of the equation (20).”

(20) Line 215: “In this paper, ……, of the belief entropy” is revise as “This paper represents a unified form of the belief entropy, which considers the scale of the frame of discernment (FOD) and the influence of the intersection between statements on uncertainty.”

(21) Line 222: “The proposed entropy……” is revise as “The proposed entropy gives a unified form of computational uncertainty, which not only considers. the scale of the FOD and the influence of the intersection between statements on uncertainly in BPA, but also solved the problem of uncertain measurement under the framework of multi-element recognition.”

Page 10:

(22) Some references of this paper are revised as follows:

[3] Charnes, A.; Cooper, W.W.; Rhodes, E. Measuring the efficiency of decision making units. European journal of operational research 1978, 2, 429–444.

[4] Bell, D.E. Regret in decision making under uncertainty. Operations research 1982, 30, 961–981.

[5] Polikar, R. Ensemble based systems in decision making. IEEE Circuits and systems magazine 2006, 6, 21–45.

[6] Edwards, W. The theory of decision making. Psychological bulletin 1954, 51, 380.

[10] Hu, M.K. Visual pattern recognition by moment invariants. IRE transactions on information theory 1962,8, 179–187.

[11] Klinke, A.; Renn, O. A New Approach to Risk Evaluation and Management: Risk-Based, Precaution-Based,and Discourse-Based Strategies 1. Risk Analysis: An International Journal 2002, 22, 1071–1094.

Page 11:

[26] Pal, N.R.; Bezdek, J.C.; Hemasinha, R. Uncertainty measures for evidential reasoning I: A review.International Journal of Approximate Reasoning 1992, 7, 165–183.

[39] Florea, M.C.; Grenier, D. Unified approach to the fusion of imperfect data? Proceedings of SPIE – The International Society for Optical Engineering 2002, 4731

We thank the reviewer for the valuable suggestions to our works. We hope that the deficiencies pointed out in the original submission are overcome in this revised version. If there are any problems in the revised version, please do not hesitate to point out, we will revise the submission according to reviewer’s suggestions.

Round 2

Reviewer 2 Report

Dear Authors,

after reading your revised version of the manuscript I have just some minor revision related to grammatical misspellings that I have highlighted in the pdf of the revised manuscript which I enclose here. Once it is done, I have no problem to recommend this work to be published.

Sincerely yours. 
